# Fungal dissemination is limited by liver macrophage filtration of the blood

Donglei Sun[1], Peng Sun[1], Hongmei Li[1,2], Mingshun Zhang ◉ [1,3], Gongguan Liu[1], Ashley B. Strickland[1], Yanli Chen[1], Yong Fu[1], Juan Xu[1,3], Mohammed Yosri[1,4], Yuchen Nan[5], Hong Zhou[3], Xiquan Zhang[2] & Meiqing Shi[1]*

Fungal dissemination into the bloodstream is a critical step leading to invasive fungal infections. Here, using intravital imaging, we show that Kupffer cells (KCs) in the liver have a prominent function in the capture of circulating *Cryptococcus neoformans* and *Candida albicans*, thereby reducing fungal dissemination to target organs. Complement C3 but not C5, and complement receptor CRIg but not CR3, are involved in capture of *C. neoformans*. Internalization of *C. neoformans* by KCs is subsequently mediated by multiple receptors, including CR3, CRIg, and scavenger receptors, which work synergistically along with C5aR signaling. Following phagocytosis, the growth of *C. neoformans* is inhibited by KCs in an IFN-γ independent manner. Thus, the liver filters disseminating fungi from circulation via KCs, providing a mechanistic explanation for the enhanced risk of cryptococcosis among individuals with liver diseases, and suggesting a therapeutic strategy to prevent fungal dissemination through enhancing KC functions.

[1] Division of Immunology, Virginia-Maryland College of Veterinary Medicine and Maryland Pathogen Research Institute, University of Maryland, College Park, USA. [2] Department of Animal Genetics, Breeding and Reproduction, College of Animal Science, South China Agricultural University, Guangzhou, China. [3] Department of Immunology, Nanjing Medical University, Nanjing, China. [4] The Regional Center for Mycology and Biotechnology, Al-Azhar University, Cairo, Egypt. [5] Department of Preventive Veterinary Medicine, College of Veterinary Medicine, Northwest A&F University, Yangling, China. *email: mshi@umd.edu

Invasive fungal diseases, caused by systematic dissemination of fungi, are a major threat to human health[1]. One of the most important fungal species causing such diseases is *Cryptococcus neoformans*. This organism causes fatal meningoencephalitis predominantly in immunocompromised patients[2,3], although the infection has been reported in apparently immunocompetent individuals[4]. It is estimated that there are one million cases of cryptococcal meningoencephalitis resulting in ~600,000 deaths every year worldwide[5]. While HIV infection is the main risk factor for cryptococcal meningoencephalitis, the use of immunosuppressive drugs[6–8] also increases patient susceptibility to this organism. Recently, increasing clinical studies have indicated that patients with cirrhosis or end-stage liver diseases are also more susceptible to brain infections with *C. neoformans*, establishing a link between liver diseases and the enhanced risk for cryptococcal meningoencephalitis[6,9–12].

*C. neoformans* initially infects the lungs. Hematogenous dissemination of the fungus from the lungs is believed to be a critical step towards meningoencephalitis[13,14]. Early mouse studies have shown that brain infection occurs following fungemia[15] and that there is a direct correlation between the magnitude of fungemia and the severity of brain infection[16]. In clinical settings, fungemia is frequently detected in patients during cryptococcosis, suggesting that fungemia is critical for the onset and persistence of cryptococcal meningoencephalitis in humans[17]. As such, intravascular clearance of disseminating *C. neoformans* from circulation likely plays an important role in preventing and ameliorating meningoencephalitis. However, it remains unknown whether a mechanism exists to actively filter disseminating *C. neoformans* out of the bloodstream.

To combat the invasion of *C. neoformans*, T cells are believed to play a critical role[18]. In general, Th1 and Th17 responses are protective while Th2 responses have been shown to be detrimental[19,20]. Thus, induction of protective Th1 responses, including enhancing IFN-γ secretion, has been used as a strategy for vaccination design against *C. neoformans*[21]. These different T cell responses differently shape the cytokine milieu which results in distinct macrophage polarization. Classically activated macrophages are suggested to control *C. neoformans* proliferation while alternatively activated macrophages are thought to be growth permissive[22–24]. The above mechanisms help explain why patients with HIV infection or undergoing immunosuppression are more susceptible to *C. neoformans* infections. However, such mechanisms cannot directly explain the correlation between cryptococcosis and liver diseases. Thus, it remains unknown why liver disease is a risk factor for cryptococcal meningoencephalitis.

The liver is the biggest internal organ receiving blood supply from both the hepatic artery and the portal vein[25]. It is estimated that every minute, 30% of the body's total volume of blood passes through the liver[25]. Kupffer cells (KCs) are liver-resident macrophages and constitute ~90% of total tissue macrophages in the body[26]. They generally reside within the lumen of the liver sinusoids, and adhere to the endothelial lining of blood vessels[25–27]. Recent studies showed that the liver is a primary surveillance organ for intravascular infections[25], and is especially important for filtering bacterial pathogens via KCs to maintain blood sterility[27–30]. It remains unknown whether the liver plays an important role in preventing fungal dissemination during invasive fungal infections (including cryptococcosis) which kill about one and a half million people every year worldwide[1,31].

We formulated a hypothesis that the liver plays a prominent role in filtering disseminating fungi out of the vasculature. In the current study, with the use of intravital microscopy (IVM), we examined the dynamic interactions between liver KCs and disseminating fungal cells and the underlying mechanisms using mouse models of infection with *C. neoformans* and *Candida albicans*. The results show that liver KCs actively capture circulating fungal cells, thus reducing their dissemination to target organs. Moreover, KCs internalize captured fungal cells and inhibit their proliferation in an IFN-γ independent manner.

## Results

**IVM of liver capture of circulating *Cryptococcu*s.** To determine whether the liver filters disseminating *C. neoformans*, we performed IVM to visualize the behavior of this organism in the liver in real-time. Following i.v. injection via the tail vein, yeast cells appeared in the liver within seconds. The yeast cells appeared to move at the same speed as the blood flow and came to a sudden stop in the liver without tethering, rolling, or crawling (Supplementary Movie 1). In a series of images taken after injection, spherical yeast cells were initially moving so fast that they were seen as luminescent streaks; in the next image (1.2 s later), the yeast cells were completely stopped (Fig. 1a, upper panel). Once stopped, a portion of yeast cells were seen to be released back into the bloodstream (Supplementary Movie 1). In another sequence of images, an organism was seen as a luminescent streak that was spherical in the prior frame; and in the third frame (2.5 s later), the yeast cells had completely disappeared (Fig. 1a, lower panel). The yeast cells were stopped in the liver sinusoids (Fig. 1b). As *C. neoformans* is nearly undetectable in the bloodstream 60 min after infection (Supplementary Fig. 1), we characterized the kinetics of yeast capture in the liver up to 50 min after infection and observed a constant increase in the number of yeast cells stopped in the liver sinusoids over the time (Fig. 1c). As KCs are the predominant liver-resident immune cells, we labeled liver KCs in vivo by i.v. injection of anti-F4/80 mAb and found that most of the yeast cells stopped in the liver were in association with KCs (Fig. 1d).

**KCs catch disseminating *Cryptococcus*.** The association of *C. neoformans* with KCs shown above raised the possibility that KCs are involved in the liver capture of circulating yeast cells. To this end, we examined the role of KCs during the capture of *C. neoformans* by using clodronate liposomes (CLL) to deplete KCs 24 h before infection with *C. neoformans*. The depletion of KCs was confirmed by confocal microscopy (Fig. 2a) and flow cytometry (Supplementary Fig. 2A). When comparing KC depleted mice with control mice after i.v. infection with *C. neoformans* using IVM, a dramatic decrease was observed in the number of yeast cells captured in the liver (Fig. 2b). This was confirmed by colony forming unit (CFU) enumeration of homogenized liver, which showed a reduction of liver CFU by one order of magnitude after KC depletion (Fig. 2c). Analysis of IVM videos showed that blood clearance of circulating *C. neoformans* was significantly impaired after KC depletion, resulting in a higher half-life of free yeast cells in the bloodstream (Fig. 2d). The impaired blood clearance of *C. neoformans* revealed by IVM was associated with higher blood CFU in the absence of KCs (Fig. 2e). IVM videos showed that, when KCs were depleted, most yeast cells were able to pass through the sinusoids smoothly without encountering obstructions (Supplementary Movie 2). Statistical analysis of IVM videos indicated that *C. neoformans* in KC depleted mice had a significantly reduced capture possibility, indicating circulating yeast cells were less likely to be stopped in the liver without KCs (Fig. 2f). Moreover, the liver showed a decreased ability to steadily hold those *C. neoformans* which had stopped, resulting in an increased escape rate in KC depleted mice (Fig. 2g). Immunohistochemistry also confirmed the reduced liver capture of *C. neoformans* in the absence of KCs in deeper sites of the liver (Supplementary Fig. 2B). The impaired clearance was associated with enhanced blood fungal burden, resulting in more deposition of *C. neoformans* in other organs (Fig. 2h). It should be noted that

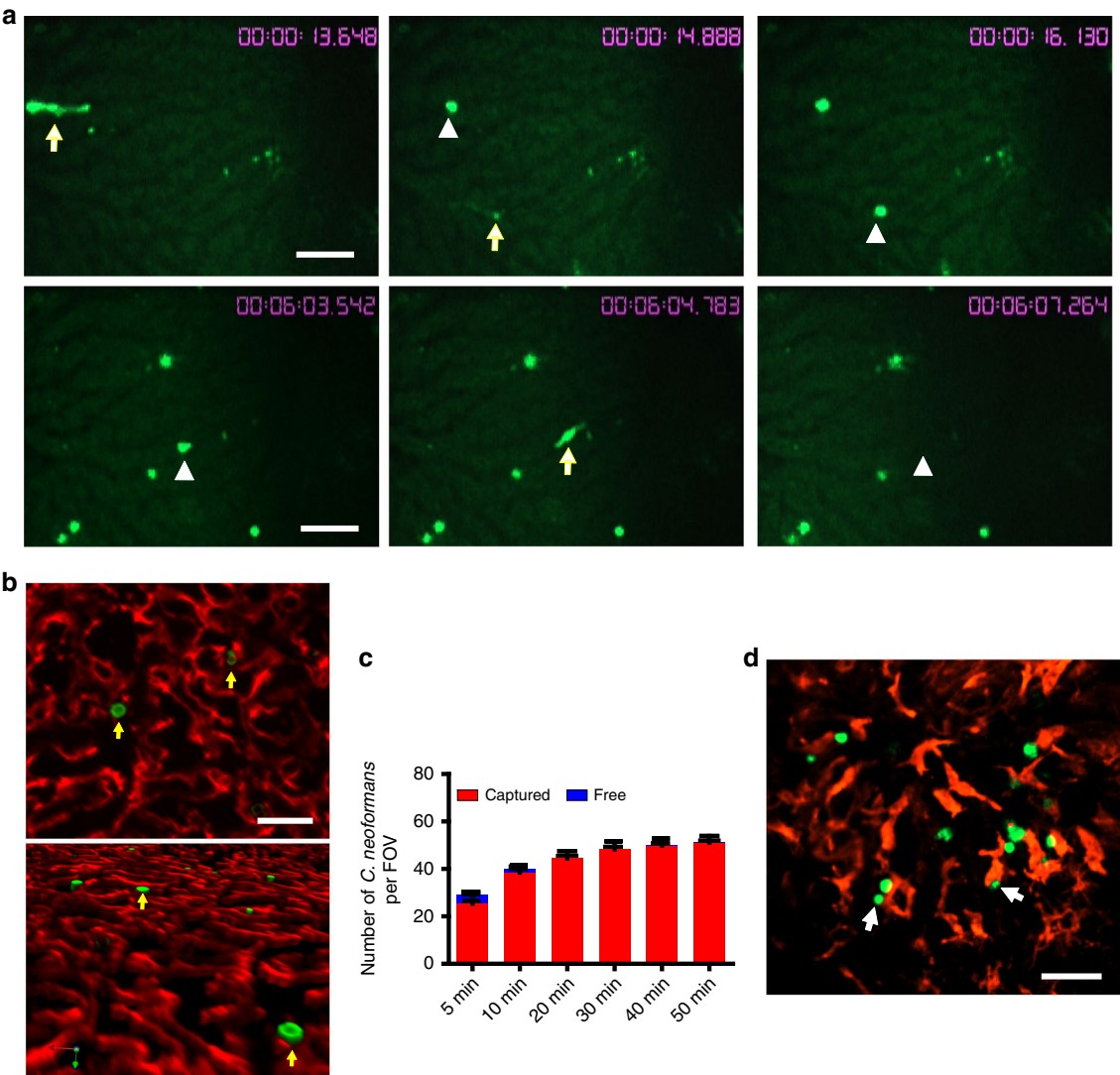

**Fig. 1** The dynamics of the capture of circulating *C. neoformans* in the liver. IVM was performed on the liver of mice ($n = 5$ mice) following i.v. injection of $100 \times 10^6$ GFP-labeled *C. neoformans* H99 via the tail vein. **a** A series of images taken by IVM showing the same field of view after injection. Time in minutes and seconds after injection is shown in the images. Upper panel: a sudden stop of the yeast cells in the liver. Arrows indicate the moving yeast cells; arrowheads indicate the same yeast cells arrested in the next frame (1.2 s later). Lower panel: release of an arrested yeast cell. Arrowhead in the left image indicates an arrested yeast cell; arrow in the middle image indicates the same yeast cells leaving in the next frame (1.2 s later); arrowhead in the right image indicates absence of the yeast cells 2.5 s later. **b** Representative IVM images showing that *C. neoformans* (green) was arrested in liver sinusoids (endothelial cells were labeled with anti-PECAM-1 mAb; red) after i.v. infection with GFP-labeled *C. neoformans*. Upper panel: 2D image; lower panel: 3D image. **c** The number of captured and free *C. neoformans* in a field of view at various time points after injection. At indicated time points, the number of yeast cells captured (being stationary for >3 s) and free yeast cells traveling in the bloodstream were counted. Data are from biologically different mice ($n = 5$ mice, one FOV per mouse). **d** PE conjugated anti-F4/80 mAb was injected into the tail vein for staining of liver KCs (red); most yeast cells (green) were associated with KCs. Scale bars: 25 μm. Data are expressed as mean ± SEM of two independent experiments. Source data are provided as a Source Data file

CLL treatment also depleted spleen macrophages (Supplementary Fig. 2C) and resulted in a reduction in spleen CFU, which indicated that spleen macrophages also help capture *C. neoformans*. Since the yeast cells are gradually released from the lung during natural infections, we also infected mice intratracheally with *C. neoformans*. Consistent with i.v. infection, mice depleted of KCs exhibited significantly reduced liver capture of *C. neoformans* and increased fungal burden in the brain (Fig. 2i). As CLL deplete not only KCs but also monocytes, we treated mice with gadolinium chloride, which was previously shown to specifically ablate KCs or to inactivate KCs[32,33]. Treatment with gadolinium chloride significantly reduced liver CFU, resulting in higher fungal burdens in the blood and other tissues (Supplementary Fig. 3). To

exclude the role of circulating monocytes, we next infected CCR2[−/−] and Nur77[−/−] mice, which lack Ly6C[hi] and Ly6C[low] monocytes in the bloodstream, respectively[34,35]. We also infected Nur77[−/−] mice depleted of Ly6C[hi] monocytes by anti-CCR2 mAb[36]. We found that loss of Ly6C[hi] and/or Ly6C[low] monocytes did not affect the fungal burdens in the liver and other tissues 3 h after infection with *C. neoformans* (Supplementary Fig. 4). Collectively, these data demonstrate that KCs are critically involved in liver capture of disseminating *C. neoformans*.

**C3 but not C5 is required for cryptococcal capture**. As *C. neoformans* is a strong activator of the complement system[37,38],

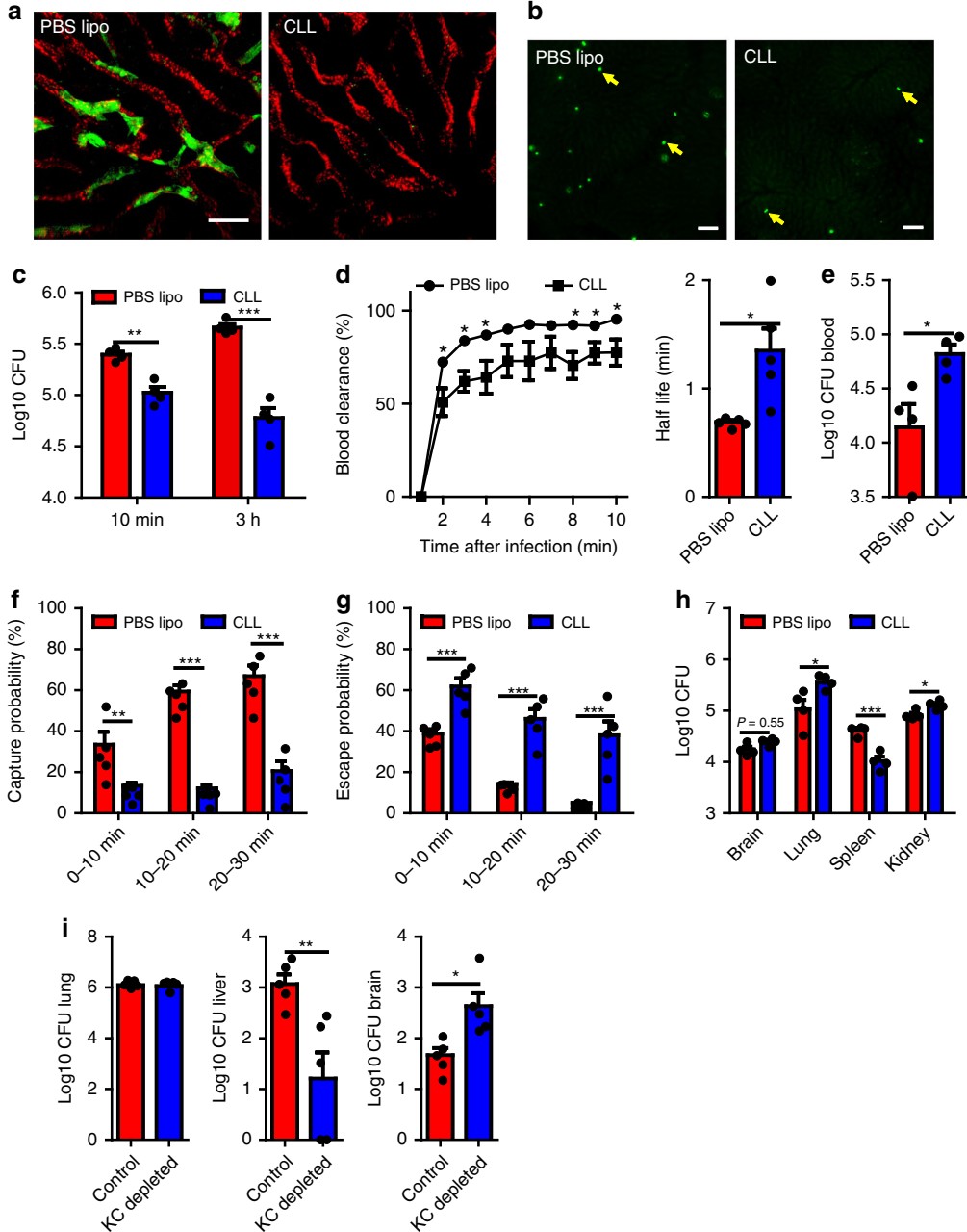

**Fig. 2** KCs are critically involved in liver capture of disseminating *C. neoformans*. **a** Confocal microscopy revealed that KCs were completely depleted 24 h after treatment with clodronate liposomes (CLL). The control mice were treated with PBS liposomes (PBS lipo). KCs (labeled by Alexa Fluor 488 anti-F4/80 mAb): green; sinusoids (labeled by Alexa Fluor 647 anti-PECAM-1 mAb): red, pseudocolor. **b** Representative IVM images showing yeast cells (green, arrows) captured in the liver of mice treated with CLL or PBS lipo 30 min after i.v. injection with $100 \times 10^6$ GFP-expressing *C. neoformans* H99. **c** *C. neoformans* ($5 \times 10^6$) was injected into the tail vein of mice ($n = 4$ mice/group) treated with CLL or PBS lipo. CFU in the liver was enumerated 10 min and 3 h post infection. **d** Vascular clearance and the half-life of circulating *C. neoformans* at various time points after injection of $100 \times 10^6$ yeast cells into the tail vein of mice ($n = 5$ mice/group) treated with CLL or PBS lipo by analysis of IVM videos. **e** Blood CFU was enumerated in mice ($n = 4$ mice/group) treated with CLL or PBS lipo 10 min after i.v. infection of $5 \times 10^6$ *C. neoformans*. **f** The possibility of circulating *C. neoformans* being captured was calculated by analysis of IVM videos from infected mice ($n = 5$ mice/group) treated with CLL or PBS lipo. **g** The possibility of trapped *C. neoformans* being released back to circulation was calculated by analysis of IVM videos from infected mice ($n = 5$ mice/group) treated with CLL or PBS lipo. **h** CFU in different organs was enumerated 3 h after i.v. injection of $5 \times 10^6$ *C. neoformans* into the tail vein of mice ($n = 4$ mice/group) treated with CLL or PBS lipo. **i** CFU of different organs was enumerated 7 days after intratracheal infection with $1 \times 10^6$ *C. neoformans* in mice ($n = 5$ mice/group) treated with CLL or PBS lipo. Data are expressed as mean ± SEM of three independent experiments. All data are from biologically distinct samples. Scale bars: 25 μm. *$p < 0.05$, **$p < 0.01$, ***$p < 0.001$, $p$ values were calculated by two-way ANOVA (**c**, **f**, **g**) or Student's *t* test (**d**, **e**, **h**, **i**). Source data are provided as a Source Data file

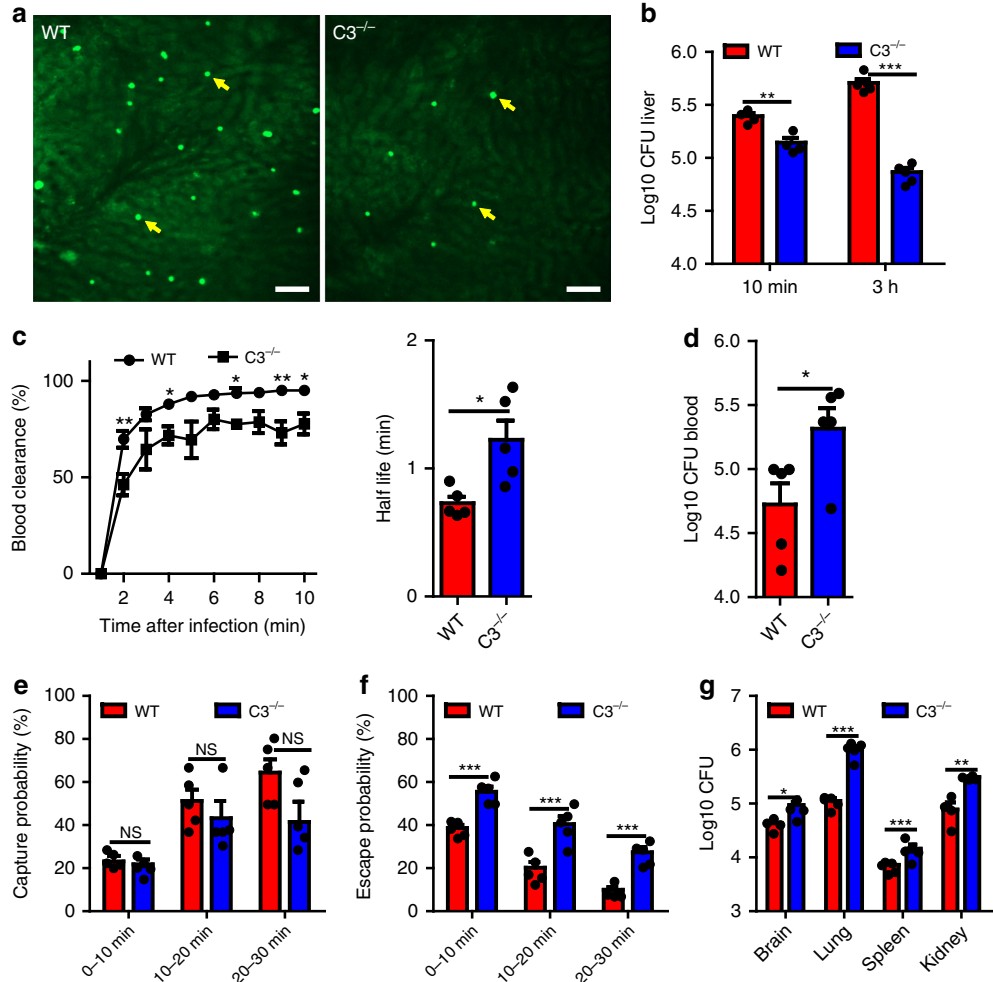

**Fig. 3** Complement is required for KCs to capture *C. neoformans*. **a** Representative IVM images showing yeast cells (green, arrows) captured in the liver of WT and C3$^{-/-}$ mice 30 min after i.v. infection with $100 \times 10^6$ GFP-expressing *C. neoformans*. **b** Liver CFU was enumerated in WT and C3$^{-/-}$ mice ($n = 4$–5 mice/group) 10 min and 3 h post i.v. infection with $5 \times 10^6$ *C. neoformans* H99. **c** Vascular clearance and the half-life of circulating *C. neoformans* at various time points after injection of $100 \times 10^6$ yeast cells into tail vein of WT and C3$^{-/-}$ mice ($n = 5$ mice/group) by analysis of IVM videos. (**d**) Blood CFU was enumerated in WT and C3$^{-/-}$ mice ($n = 5$ mice/group) 10 min post i.v. infection of $5 \times 10^6$ *C. neoformans*. **e** The possibility of circulating *C. neoformans* being captured was calculated by analysis of IVM videos from infected WT and C3$^{-/-}$ mice ($n = 5$ mice/group). **f** The possibility of trapped *C. neoformans* being released back to circulation was calculated by analysis of IVM videos from infected WT and C3$^{-/-}$ mice ($n = 5$ mice/group). **g** CFU in different organs was enumerated 3 h after injection of $5 \times 10^6$ *C. neoformans* into the tail vein of WT and C3$^{-/-}$ mice ($n = 4$–5 mice/group). Data are expressed as mean ± SEM of 3 independent experiments. All data are from biologically distinct samples. Scale bars: 25 μm. *$p < 0.05$, **$p < 0.01$, ***$p < 0.001$. NS: not significant. $p$ values were calculated via two-way ANOVA (**e**, **f**) or Student's $t$ test (**b**, **c**, **d**, **g**). Source data are provided as a Source Data file

we next examined the role of complement during liver capture of *C. neoformans*. IVM was performed on the livers of WT and C3$^{-/-}$ mice following infection with *C. neoformans*. Compared to WT mice, C3$^{-/-}$ mice exhibited significantly fewer yeast cells in the liver following infection (Fig. 3a, Supplementary Movie 3), and most yeast cells were released back into the bloodstream following a transient stop (Supplementary Movie 3). A lower liver fungal burden confirmed the reduced liver capture of circulating yeast cells in C3$^{-/-}$ mice (Fig. 3b). The reduced capture ability in the liver resulted in impaired blood clearance and an increased half-life of circulating yeast cells in C3$^{-/-}$ mice (Fig. 3c), which was confirmed by the enhanced blood CFU in C3$^{-/-}$ mice post infection (Fig. 3d). In contrast to KC depletion, analysis of IVM videos showed that the decrease of liver CFU in C3$^{-/-}$ mice was not due to the initial stop of *C. neoformans*. In fact, there was no significant difference in the capture probability of *C. neoformans* between WT and C3$^{-/-}$ mice (Fig. 3e). Instead, C3$^{-/-}$ mice showed impaired ability to stably hold the stopped *C. neoformans*,

i.e., more stopped yeast cells were released back into circulation in C3$^{-/-}$ mice compared to WT control (Fig. 3f). The reduced liver capture of yeast cells in C3$^{-/-}$ mice was associated with significantly higher fungal burden in other organs (Fig. 3g). Thus, although C3 was not required for the initial stop of *C. neoformans*, it is essential for *C. neoformans* to be retained in the liver.

To further confirm the role of complement during fungal capture, we incubated *C. neoformans* with fresh serum from either WT or C3$^{-/-}$ mice, and injected an equal number of WT or C3$^{-/-}$ serum treated yeast cells into C3$^{-/-}$ mice to determine whether opsonization could overcome the deficiency of liver capture of the organism. Incubation with WT, but not C3$^{-/-}$ mouse serum, resulted in a complete coating of C3b/iC3b on fungal surfaces (Supplementary Fig. 5A), and this coating of C3b/iC3b significantly rescued the liver's capture ability in C3$^{-/-}$ mice (Supplementary Fig. 5B). The complement pathway is known to be heat liable; as a result, the recovery of capture ability in C3$^{-/-}$ mice was lost when WT serum was heat inactivated

(Supplementary Fig. 5C). In addition to C3b/iC3b, C5b and C6-C9 are also deposited on microbial surfaces during opsonization. To determine the role of these late complement activation components during liver capture of *C. neoformans*, WT and C5$^{-/-}$ mice were i.v. infected with *C. neoformans*. No significant difference was observed in the liver CFU between infected WT and C5$^{-/-}$ mice, indicating that C5b and late activation components were not required for liver capture of *C. neoformans* (Supplementary Fig. 5D). Taken together, these experiments demonstrate that C3 but not C5 is required for KCs to capture circulating *C. neoformans*.

**CRIg but not CR3 is required for cryptococcal capture**. Complement receptor 3 (CR3 or CD11b/CD18) is the most well characterized receptor for pathogen recognition. However, KCs only express low levels of CD11b, compared to other phagocytes such as neutrophils (Supplementary Fig. 5E). To determine whether the low expression of CR3 contributes to liver capture of *C. neoformans*, liver fungal burdens of WT and CD11b$^{-/-}$ mice were compared following i.v. infection. The result indicated that CR3 plays a negligible role in liver capture of *C. neoformans* (Supplementary Fig. 5F).

In addition to CR3, a unique complement receptor, CRIg, was recently identified as being expressed on KCs[39]. Using immunohistochemistry and flow cytometry, we confirmed CRIg expression on KCs (Fig. 4a, b) but not spleen macrophages (Supplementary Fig. 5G). Since the gene encoding CRIg is located on the X chromosome, we also checked expression levels for male and female mice using flow cytometry. We did not detect significant differences in CRIg expression between male and female mice (Fig. 4b), which indicated that this gene does not escape X-inactivation. To determine the role of CRIg in liver capture of *C. neoformans*, WT and CRIg$^{-/-}$ mice were i.v. infected with *C. neoformans*. Compared to infected WT mice, infected CRIg$^{-/-}$ mice displayed a significantly lower fungal burden in the liver (Fig. 4c). This was associated with enhanced blood CFU (Fig. 4d), which led to higher fungal burdens in other organs (Fig. 4e). IVM revealed dramatically reduced capture of *C. neoformans* in the livers of CRIg$^{-/-}$ mice compared to WT mice (Fig. 4f). Analysis of IVM videos showed impaired blood clearance and an enhanced half-life of circulating yeast cells in CRIg$^{-/-}$ mice (Fig. 4g). In consistence with C3$^{-/-}$ mice, there was no significant difference in capture probability of *C. neoformans* between CRIg$^{-/-}$ and WT mice (Fig. 4h). However, CRIg$^{-/-}$ mice showed a significantly higher escape probability (Fig. 4i), implying that more yeast cells were released back into the bloodstream after a transient stop. Together, these results demonstrate that CRIg but not CR3 is critically involved in KC mediated capture of circulating yeast cells.

**Uptake of *Cryptococcus* by KCs involves multiple receptors**. KCs are professional phagocytes. Following capture of *C. neoformans*, most of the yeast cells were internalized by KCs as shown by IVM (Fig. 5a; Supplementary Movie 4). To reveal the mechanisms underlying phagocytosis of *C. neoformans*, we developed a system, based on published literature, to distinguish yeast cells outside KCs from those inside KCs in mice infected with GFP-expressing *C. neoformans* by i.v. injection of the fungal specific dye Uvitex 2B[40,41]. GFP-expressing yeasts outside phagocytes can be stained by Uvitex 2B while those phagocytized cannot (Supplementary Fig. 6A). As shown in Fig. 5b and Supplementary Fig. 6B, *C. neoformans* captured in the liver of WT mice was rapidly engulfed, with more than 90% of the yeast cells located intracellularly in as short as 15 min after infection. By

contrast, only half of the yeast cells were engulfed in C3$^{-/-}$ mice 15 min after infection. However, most of the yeast cells were eventually phagocytized in C3$^{-/-}$ mice at a later time point (Fig. 5b). Nevertheless, C3$^{-/-}$ mice showed significantly reduced phagocytosis rates at all-time points examined (Fig. 5b), indicating a role of C3 in KC phagocytosis of *C. neoformans*. Interestingly, C5$^{-/-}$ mice also exhibited delayed and impaired phagocytosis of *C. neoformans* in general compared to WT mice (Fig. 5b). Thus, phagocytosis of captured *C. neoformans* by liver KCs is mediated by C3, with involvement of C5.

To identify the receptors involved in phagocytosis of *C. neoformans* on KCs, we compared the phagocytosis efficiency of WT mice with CD11b$^{-/-}$ (CR3 deficient), CRIg$^{-/-}$ mice, and double deficient mice (CRIg$^{-/-}$ mice treated with CD11b blocking mAb). Interestingly, single deficiency of either CD11b or CRIg did not affect phagocytosis efficiency. However, the deficiency of both CRIg and CD11b significantly reduced the phagocytosis rate (Fig. 5e), indicating functional redundancy between CR3 and CRIg on KCs. In addition to CR3 and CRIg, KCs express high levels of C5aR as revealed by flow cytometry (Fig. 5c); a blockade for C5aR using blocking mAb significantly reduced the phagocytosis rate in WT mice (Fig. 5d), indicating a critical involvement of C5aR signaling. Given that some of the yeast cells could eventually be phagocytized in the liver of C3$^{-/-}$ mice (Fig. 5b), it is conceivable that other receptors also contribute to phagocytosis independent of complement. Accordingly, we found that inhibition of scavenger receptors by Poly I significantly inhibited phagocytosis especially in C3$^{-/-}$ mice which do not have complement-mediated phagocytosis (Fig. 5f, g). Collectively, these results suggest that phagocytosis of *C. neoformans* in the liver under shear stress is mediated by both complement-dependent and -independent pathways involving multiple receptors including CR3, CRIg, and scavenger receptors, and that the process can be significantly enhanced by the presence of C5aR signaling.

**KCs inhibit cryptococcal growth after phagocytosis**. The ability of *C. neoformans* to grow within macrophages prompted us to study the fate of this organism following phagocytosis by KCs. Interestingly, the vast majority of yeast cells remained as single-cell colonies in the liver 3 days after infection, whereas more than 90% of yeast cells trapped in other organs were proliferating and forming fungal clusters (Fig. 6a, b). As we have shown above, most of the yeast cells captured in the liver were engulfed by KCs, this finding raised the possibility that KCs inhibit *C. neoformans* growth in the liver. To this end, we depleted KCs using CLL before infection and found that *C. neoformans* proliferated in the liver as it did in other organs (Fig. 6c). In livers depleted of KCs, the majority of yeast cells started to grow and formed clusters, which is in contrast to the normal liver where fungal growth was suppressed (Fig. 6d).

IFN-γ can classically activate macrophages, resulting in killing or growth inhibition of *C. neoformans* within macrophages[24]. Invariant natural killer T cells (iNKT cells) are abundant in the liver and are known to produce IFN-γ following activation. Interestingly, IVM on the liver of CXCR6$^{gfp/+}$ mice revealed that clusters of iNKT cells were formed in the liver of mice following i.v. infection with *C. neoformans*; but this was not seen in the liver of naïve mice (Fig. 6e, f). Clusters of iNKT cells have also been seen in the liver following infection with the bacterial pathogen *Borrelia burgdorferi*, which is linked to iNKT cell activation and IFN-γ secretion[42]. In consistence with the published study[42], we detected IFN-γ production in the liver following i.v. infection with *C. neoformans* (Fig. 6g). To determine whether IFN-γ has

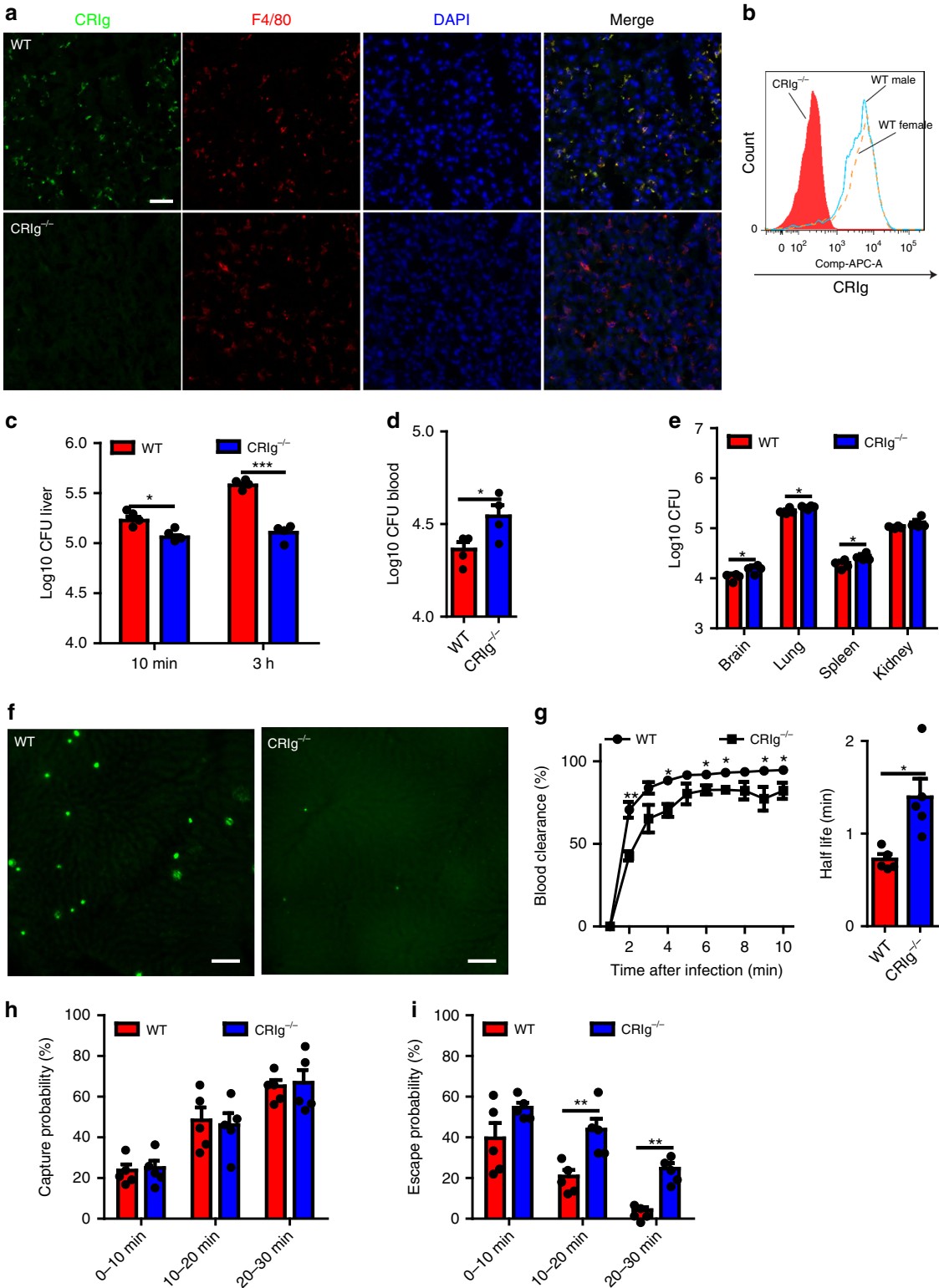

the potential to activate KCs, we confirmed that KCs express IFN-γR via flow cytometry (Fig. 6h). Surprisingly, IFN-γR⁻/⁻ mice did not show any impairment in inhibition of fungal growth in the liver as determined by enumerating the yeast cells per colony (Fig. 6i). This was confirmed by the evidence that there was no significant difference in the liver CFU between infected WT and IFN-γR⁻/⁻ mice (Fig. 6j). Thus, KCs inhibit *C. neoformans* growth following phagocytosis, in a manner which is independent of IFN-γR signaling.

**KCs catch circulating *Candida* and inhibit hyphal growth**. Having demonstrated the critical role of the liver in filtering disseminating *C. neoformans* via KCs, we next examined whether liver KCs also capture circulating *C. albicans*, another important fungal pathogen responsible for invasive candidiasis[43]. Depletion of KCs using CLL dramatically reduced the liver fungal burden (Fig. 7a), leading to more deposition of yeast cells in the target organ, the kidney, following i.v. infection with *C. albicans* (Fig. 7b). Importantly, almost all yeast cells developed hyphae in

**Fig. 4** CRIg is the receptor on KCs mediating the capture of *C. neoformans*. **a** Immunofluorescence staining showing the expression of CRIg (green) on KCs (red) of WT mice (upper panel) but not CRIg$^{-/-}$ mice (lower panel). The nuclei were stained with DAPI (blue). **b** The expression of CRIg on KCs of WT (male and female) but not CRIg$^{-/-}$ mice (gated on CD45$^+$F4/80$^+$CD11b$^{int}$ population). **c** Liver CFU was enumerated in WT and CRIg$^{-/-}$ mice ($n = 4$ mice/group) 10 min and 3 h after i.v. infection with $5 \times 10^6$ *C. neoformans*. **d** Blood CFU was enumerated in WT and CRIg$^{-/-}$ mice ($n = 4$ mice/group) 10 min post i.v. infection of $5 \times 10^6$ *C. neoformans*. **e** CFU of different organs was enumerated in WT and CRIg$^{-/-}$ mice ($n = 4$ mice/group) 3 h after i.v. infection of $5 \times 10^6$ *C. neoformans*. **f** Representative IVM images showing yeast cells (green) captured in the liver of WT and CRIg$^{-/-}$ mice ($n = 5$ mice/group) 30 min after i.v. infection with $100 \times 10^6$ GFP-labeled *C. neoformans*. **g** Vascular clearance and half-life of circulating *C. neoformans* at various time points after injection of $100 \times 10^6$ yeast cells into tail vein of WT and CRIg$^{-/-}$ mice ($n = 5$ mice/group) by analysis of IVM videos. **h** The possibility of circulating *C. neoformans* being captured was calculated by analysis of IVM videos from infected WT and CRIg$^{-/-}$ mice ($n = 5$ mice/group). **i** The possibility of trapped *C. neoformans* being released back to circulation was calculated by analysis of IVM videos from infected WT and CRIg$^{-/-}$ mice ($n = 5$ mice/group). Scale bars: 25 μm. Data are expressed as mean ± SEM of two independent experiments. All data are from biologically distinct samples. *$p <$ 0.05, **$p < 0.01$, ***$p < 0.001$. $p$ values were calculated via two-way ANOVA (**c**, **h**, **i**) or Student's *t* test (**d**, **e**, **g**). Source data are provided as a Source Data file

the liver depleted of KCs but not in the normal liver (Fig. 7c, d), implying that the yeast cells were engulfed by KCs and that KCs inhibited hyphal growth. In contrast, treatment with CLL did not affect the hyphal growth in the kidney of mice infected with *C. albicans* (Fig. 7e, f). Thus, in addition to *C. neoformans*, KCs also mediate liver capture of circulating *C. albicans* and inhibit its hyphal growth.

## Discussion

Approximately 1.2 billion people suffer from fungal diseases worldwide[43]. The most serious manifestation occurs when pathogenic fungi disseminate from the initial infection sites into the blood, where they are able to cause invasive fungal infections, accounting annually for one and a half million deaths worldwide[1,31]. Although it is known that blood dissemination is a critical step for the onset of invasive fungal diseases, the mechanisms to prevent dissemination have been poorly understood. This is mainly because fungal dissemination via the host circulatory system is a dynamic process, and investigation of mechanisms to prevent dissemination requires dynamic observation approaches. In this study, with the use of IVM, we examined the dynamic interactions of KCs with circulating *C. neoformans* under shear forces and flow conditions in vivo, demonstrating that the liver plays a prominent role in filtering disseminating *C. neoformans* out of vasculature via KCs in both i.v. and pulmonary infection models, leading to less fungal deposition in the target organ. Importantly, liver KCs also catch *C. albicans* circulating in the bloodstream, reducing kidney infection. Thus, our studies reveal that a mechanism exists in the liver of hosts to actively filter disseminating fungal pathogens out of circulation during invasive fungal infections.

*C. neoformans* is an opportunistic fungal pathogen which causes meningoencephalitis primarily in immunocompromised individuals[44,45]. Alveolar macrophages are believed to play a vital role in the initial host response after *C. neoformans* infection[46]. HIV infection and immunosuppressive therapies are major risk factors for the development of brain infection, because individuals infected with HIV or treated with immunosuppressive drugs are unable to contain *C. neoformans* in the lung and the organism disseminates to the brain via the bloodstream[13,14]. Recently, accumulating clinical evidence has established a strong correlation between liver diseases including cirrhosis and cryptococcal meningoencephalitis, suggesting that liver disease is another independent risk factor for cryptococcal meningoencephalitis[6,10,47]. However, it remains completely unknown as to how liver diseases link to cryptococcal meningoencephalitis. One major contribution of the current study to the field is that the identification of the critical role of the liver in filtering disseminating *C. neoformans* via KCs provides a

mechanistic explanation for the enhanced susceptibility of individuals with liver diseases to cryptococcal infection. In line with this explanation, liver cirrhosis patients show an impaired ability to clear blood bacteria with prolonged bacteremia persistence[48] and have a higher risk of dying from bacterial infections based on existing medical data on infectious risks and related preventive and treatment data for cirrhotic patients[49].

In the past decade, extensive studies have been performed focusing on the capture of bacterial pathogens by KCs in the liver[27,29,39,42,50–53]. Nevertheless, the underlying mechanisms have not been elucidated until recently. Broadley et al. recently reported that a dual-track mechanism consisting of parallel fast and slow pathways exists for KCs to capture a broad panel of Gram-positive and Gram-negative bacteria[29]. The fast clearance of platelet-free bacteria is mediated by scavenger receptors while platelet binding shifts bacteria from fast to a slow clearance pathway mediated by the complement receptor CRIg[29]. In the absence of complement, circulating bacteria are captured by KCs though scavenger receptors but not CRIg[29]. This explains why there was no difference in liver capture of *Staphylococcus aureus* between C3$^{-/-}$ mice and C3$^{-/-}$CRIg$^{-/-}$ mice, despite the difference in liver capture of *S. aureus* between CRIg$^{-/-}$ mice and wild-type mice as reported earlier by Helmy et al.[39]. Indeed, CRIg expressed on KCs functions as a pattern recognition receptor to directly bind circulating bacteria through lipoteichoic acid recognition[27]. Although CRIg was involved in slow clearance of bacteria as reported recently by Broadley et al.[29], it could not be ruled out that CRIg bound bacterial lipoteichoic acid rather than C3b/iC3b in this experimental setting[29]. Therefore, although CRIg was originally identified as a macrophage complement receptor, whether CRIg binds bloodborne pathogens through interacting with complement in vivo under flow conditions has not been fully addressed.

Interestingly, although opsonization by complement facilitates bacterial binding to macrophages under static conditions[54,55], complement is dispensable for KCs to capture circulating bacteria such as *Listeria monocytogenes* and *S. aureus* under flow conditions[27,29,52]. In the current study, IVM revealed that *C. neoformans* was suddenly stopped in the liver sinusoids in the same manner as it was seen to be mechanically trapped in the capillary bed of the brain[43,56]. Accordingly, deficiency of C3 or CRIg did not affect the initial stop of *C. neoformans* at all. However, C3 and CRIg are required for the yeast cells to be retained in the sinusoids, indicating that both C3 and CRIg are involved in KC-mediated capture of *C. neoformans*. Among all identified complement receptors, KCs only express CR3 and CRIg[39]. Given that deficiency of CR3 did not affect the capture of *C. neoformans*, it is highly likely that CRIg is involved in C3-dependent KC-mediated capture of disseminating *C. neoformans*, unless C3 acts via a currently unidentified receptor on KCs. In

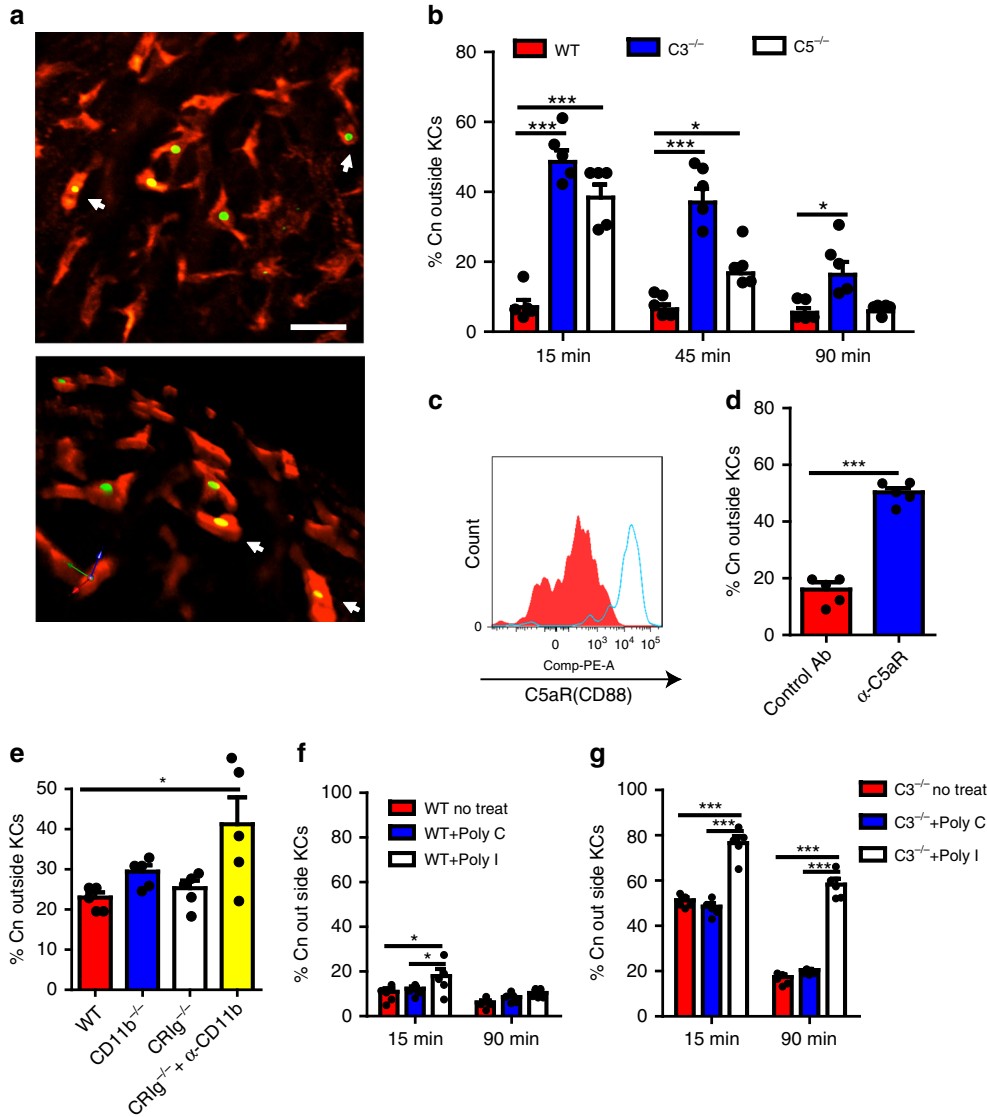

**Fig. 5** *C. neoformans* is rapidly engulfed by KCs through multiple receptors. **a** IVM on the liver showing phagocytosis of *C. neoformans* (green) by KCs (red) 1 h post i.v. infection of $100 \times 10^6$ GFP-labeled *C. neoformans*. Upper panel: 2D image; lower panel: 3D reconstructive image of the upper panel. **b** The percentage of *C. neoformans* outside KCs in the liver was determined by in vivo staining of fungi by Uvitex 2B in WT, C3$^{-/-}$, and C5$^{-/-}$ mice ($n = 5$ mice/group) at various time points following i.v. infection with $20 \times 10^6$ GFP-expressing *C. neoformans* H99. **c** The expression of C5aR on KCs (gated on CD45$^+$F4/80$^+$CD11b$^{int}$ population) was analyzed by flow cytometry. **d** The role of C5aR signaling on phagocytosis of *C. neoformans* by KCs in the liver was determined by in vivo staining of fungi by Uvitex 2B in mice ($n = 5$ mice/group) treated with anti-C5aR blocking antibody or control antibody 10 min after i.v. infection with $20 \times 10^6$ *C. neoformans*. **e** The percentage of *C. neoformans* outside KCs was determined in the liver of WT, CD11b$^{-/-}$ (CR3 deficient), CRIg$^{-/-}$ mice, and CRIg$^{-/-}$ mice treated with anti-CD11b blocking antibody ($n = 5$ mice/group) 10 min post i.v. infection of $20 \times 10^6$ GFP-expressing *C. neoformans*. **f, g** To block scavenger receptors, WT and C3$^{-/-}$ mice ($n = 5$ mice/group) were i.v. injected with 400 μg Poly(I) or 400 μg Poly(C) as control, followed by i.v. infection of $20 \times 10^6$ GFP-expressing *C. neoformans* 2 min later. The percentage of *C. neoformans* outside KCs was determined in the liver of WT mice (**f**) and C3$^{-/-}$ mice (**g**) 15 and 90 min after infection. Scale bar: 25 μm. Data are expressed as mean ± SEM of two independent experiments. All data are from biologically distinct samples. *$p < 0.05$, ***$p < 0.001$. $p$ values were calculated via two-way ANOVA (**b**, **f**, **g**), one-way ANOVA followed by Tukey's post hoc test (**e**) or Student's $t$ test (**d**). Source data are provided as a Source Data file

addition, by contrast to bacteria[29], we found that platelets were not involved in the KC-mediated capture of *C. neoformans* (Supplementary Fig. 7). Thus, although KCs mediate both fungal and bacterial capture, the underlying mechanisms are different, i.e., capture of fungal pathogens such as *C. neoformans* is both CRIg and C3 dependent while capture of bacterial pathogens such as *S. aureus* and *L. monocytogenes* is CRIg dependent but C3 independent. The ability to activate complement and the size differences among the microorganisms may account for the difference. *C. neoformans* can efficiently activate complement,

leading to a rapid and heavy deposition of C3b/iC3b on the surface of yeast cells[38]. By contrast, bacteria such as *S. aureus* develop an evasion strategy to inhibit C3b deposition onto their surfaces[57,58]. In addition, *C. neoformans* is much bigger than bacteria, resulting in a mechanical trapping of the yeast cells in the sinusoids. The initial transient stop provides a static condition which may allow for interactions between CRIg and the C3b/iC3b deposited on the yeast cells to occur[39].

Notably, *C. neoformans* was rapidly engulfed by KCs following capture. By contrast, *L. monocytogenes* were bound extracellularly

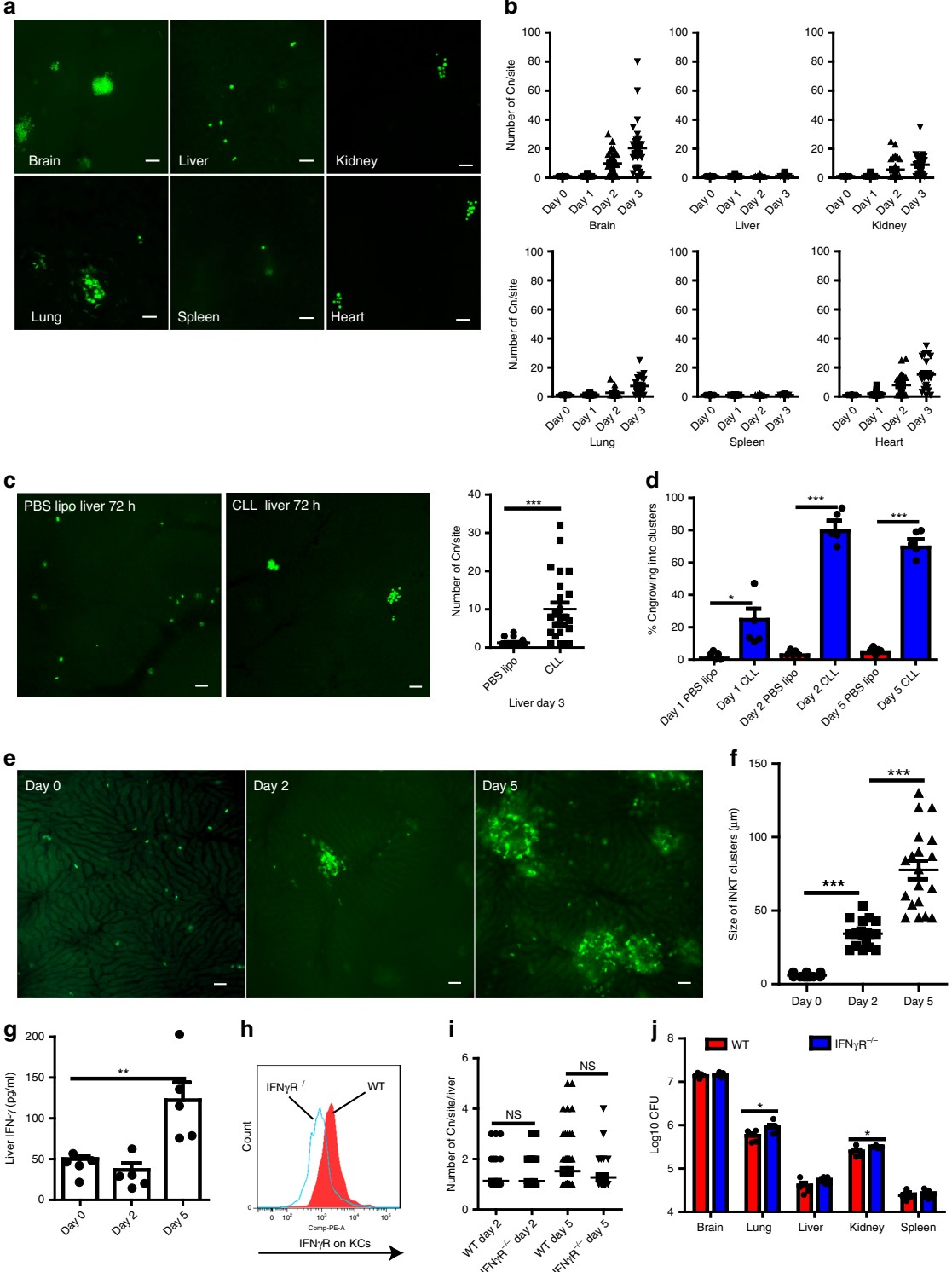

to KCs without being phagocytized[59]. We demonstrated that C3, together with CR3 and CRIg, is involved in KC-mediated phagocytosis of *C. neoformans*, a process facilitated by C5aR signaling. It remains to be elucidated as to how C5aR signaling enhances phagocytosis. Interestingly, scavenger receptors are also involved in KC-mediated phagocytosis of *C. neoformans*, which is consistent with earlier studies showing the uptake of bacteria by KCs through scavenger receptors[50]. It is worth to note that complement-mediated phagocytosis of *C. neoformans* nearly overshadowed the effect of scavenger receptors and that the effect of these receptors become pronounced only in the absence of complement.

In sharp contrast to alveolar macrophages, which promote the intracellular growth of *C. neoformans*[46], we reveal that KCs inhibit the growth of ingested yeast cells. Thus, KCs are not only critical to filter *C. neoformans* out of vasculature, but also limit fungal growth, reducing the risk of secondary dissemination of the yeast cells, demonstrating significant contribution of KCs to

**Fig. 6** KCs suppress the growth of *C. neoformans* in an IFN-γ-independent manner. **a** Representative images showing the growth of *C. neoformans* in different organs 3 days after i.v. infection with $20 \times 10^6$ GFP-labeled *C. neoformans*. **b** The number of *C. neoformans* per colony at various time points after infection. **c** Mice were treated with CLL or PBS lipo and 24 h later i.v. infected with $20 \times 10^6$ GFP-labeled *C. neoformans*. The yeast cells at each colony were counted in the liver 3 days post infection. Left panel: representative images showing the growth of *C. neoformans*; right panel: the number of *C. neoformans* per colony. **d** The percentage of *C. neoformans* growing in the liver of mice treated with CLL or PBS lipo following i.v. infection of $20 \times 10^6$ GFP-labeled *C. neoformans*. **e** Representative IVM images showing clusters of iNKT cells in the liver of CXCR6$^{gfp/+}$ mice 0, 2, and 5 days post i.v. infection with $20 \times 10^6$ *C. neoformans*. **f** The size of iNKT cell clusters in the liver of CXCR6$^{gfp/+}$ mice 0, 2, and 5 days post i.v. infection with $20 \times 10^6$ *C. neoformans*. **g** IFN-γ amounts in the liver homogenates of mice i.v. infected with $20 \times 10^6$ *C. neoformans*. **h** The expression of IFN-γR on KCs of WT but not IFN-γR$^{-/-}$ mice (gated CD45$^+$F4/80$^+$CD11b$^{int}$ population). **i** WT and IFN-γR$^{-/-}$ mice were i.v. infected with $20 \times 10^6$ GFP-labeled *C. neoformans*. The fungal growth was assessed by counting the yeast cells at each colony in the liver 2 and 5 days after infection. **j** The fungal burdens in different organs of WT and IFN-γR$^{-/-}$ mice 5 days following i.v. infection with $1 \times 10^6$ *C. neoformans*. Data from different field of views of mice ($n = 5$ mice/group) were pooled together (**b**, **c**, **f**, **i**). Data are from biologically distinct samples ($n = 5$ mice/group, **d**, **g**, **j**). Scale bars: 25 μm. Data are expressed as mean ± SEM of two independent experiments. $^*p < 0.05$, $^{**}p < 0.01$, $^{***}p < 0.001$. NS: not significant. *p* values were calculated by one-way ANOVA followed by Tukey's post hoc test (**d**, **f**, **g**, **i**) or Student's *t* test (**c**, **j**). Source data are provided as a Source Data file

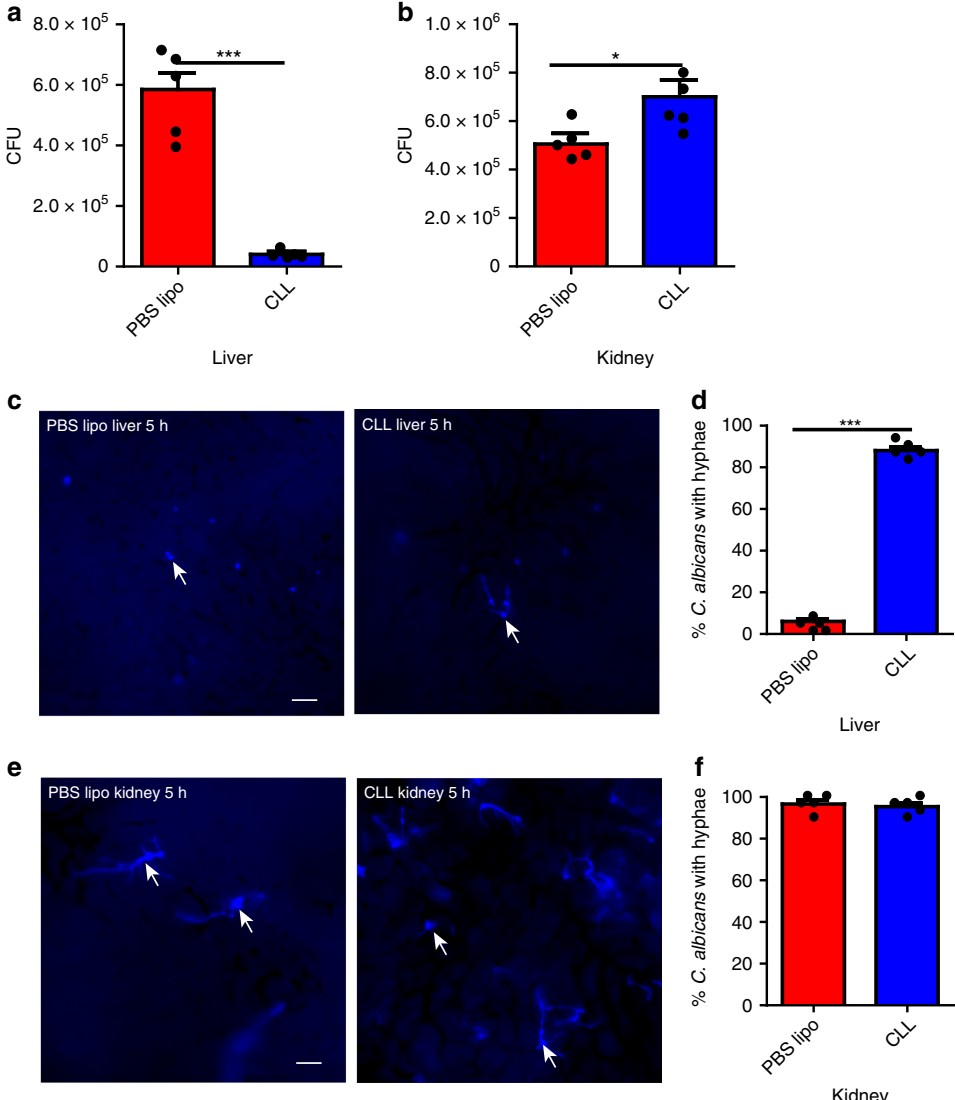

**Fig. 7** Liver KCs catch circulating *C. albicans* and inhibit its hyphal growth. **a**, **b** Mice ($n = 5$ mice/group) were treated with CLL or PBS lipo as control to deplete KCs and 24 h later i.v. infected with $5 \times 10^6$ *C. albicans*. CFU was enumerated 3 h post infection in the liver (**a**) and kidney (**b**). **c**, **d** Mice ($n = 5$ mice/group) were treated with CLL or PBS lipo as control to deplete KCs and 24 h later i.v. infected with $20 \times 10^6$ Uvitex 2B-labeled *C. albicans*. IVM was performed on the liver 5 h after infection to image the fungi. Representative images showing *C. albicans* in the liver (**c**). The percentage of *C. albican* with hyphae was determined by IVM (**d**). **e**, **f** Mice ($n = 5$ mice/group) were treated with CLL or PBS lipo as control to deplete KCs and 24 h later i.v. infected with $20 \times 10^6$ Uvitex 2B-labeled *C. albicans*. Kidney was excised 5 h post infection and immediately visualized under microscope. Representative images showing *C. albicans* in the kidney (**e**); The percentage of *C. albicans* with hyphae was determined (**f**). Scale bars: 25 μm. Data are expressed as mean ± SEM of two independent experiments. All data are from biologically distinct samples. $^*p < 0.05$, $^{***}p < 0.001$. *p* values were calculated by Student's *t* test (**a**, **b**, **d**, **f**). Source data are provided as a Source Data file

preventing fungal dissemination on multiple levels. Interestingly, phagocytosis of *C. neoformans* by KCs leads to clustering of iNKT cells, which has been seen in the liver following phagocytosis of *B. burgdorferi* by KCs and is indicative of iNKT cell activation[42]. IFN-γ detected in the liver following phagocytosis of *C. neoformans* was likely secreted by the activated iNKT cells. Surprisingly, absence of IFN-γ-signaling did not affect the growth inhibition of *C. neoformans* within KCs, demonstrating KCs inhibit fungal growth in an IFN-γ-independent manner. This result is of interest because it implies that enhancing liver macrophage recruitment can be a potential strategy to prevent dissemination of *C. neoformans*, even for those who are infected with HIV and have an impaired secretion of IFN-γ due to loss of $CD4^+$ T cells.

In conclusion, with the use of IVM we reveal that a mechanism exists to actively filter disseminating fungal pathogens via liver KCs, providing a mechanistic explanation for the higher risk of cryptococcal meningoencephalitis to those suffering from liver diseases. We further demonstrate that C3 is critically involved in the capture of yeast cells through CRIg signaling, a mechanism that has not been reported in KC-mediated capture of bacterial pathogens. Following capture, fungal pathogens are rapidly engulfed by KCs via multiple receptors, working synergistically, which is driven by C5aR signaling. Interestingly, KCs inhibit the growth of ingested yeast cells in an IFN-γ independent manner. Thus, KCs play a prominent role in preventing fungal dissemination via multiple mechanisms and enhancing KC functions may ameliorate invasive fungal infections in clinical settings.

## Methods

**Animals**. Wild type C57BL/6 mice were purchased from the National Cancer Institute (NCI). All knockout mice including $C3^{-/-}$ (Catalog No. 003641), $C5^{-/-}$ (Catalog No. 000461), $CD11b^{-/-}$ (Catalog No. 003991), $CRIg^{-/-}$ (Catalog No. 024408), $IFNγR^{-/-}$ (Catalog No. 003288), and $CXCR6^{gfp/gfp}$ (Catalog No. 005693) mice were purchased from The Jackson Laboratory and bred in the animal facilities of University of Maryland, College Park. Mice were always maintained in specific pathogen free individually ventilated cages. For all experiments, 8- to 12- week old mice with body weight ranging from 18 to 22 g were used. The authors affirm that the study complied with all relevant ethical regulations for animal testing and research. The animal study protocol was approved by the Institutional Animal Care and Use Committee (IACUC) of the University of Maryland, College Park.

**Fungal strains and culture**. Encapsulated *C. neoformans* H99 strain (serotype A) was used in this study. To visualize the fungi under IVM, GFP-expressing *C. neoformans* H99, a gift of Dr. Robin May (University of Birmingham), was used in the experiments. For experiments related to *Candida albicans*, the wild-type strain (SC5314) was used. The fungi were cultured to exponential phase at 32 °C with gentle rotation in an orbital shaker (180 rpm) in Sabouraud dextrose broth (BD) and harvested by centrifugation at $400 \times g$ for 5 min. After washing with PBS twice, the fungal concentration was determined using a hemocytometer.

**Infections and treatments**. In most experiments, mice were infected with $5 \times 10^6$, $20 \times 10^6$, or $100 \times 10^6$ *C. neoformans* or *C. albicans* via the tail vein. In some experiments, mice were intratracheally infected with $1 \times 10^6$ *C. neoformans* in 50 μl sterile PBS via a midline neck incision under anesthesia. Depletion of KCs was achieved by i.v. treatment with 200 μl of CLL 24 h before infection. The depletion of KCs can be maintained for as long as 1 week after a single treatment. Platelets were depleted by treating mice with 100 μg of anti-CD41 mAb (Catalog No. 133910, Biolegend). For blocking scavenger receptors, mice were i.v. injected with 400 μg of polyinosinic acid (poly(I)) or polycytidylic acid (poly(C)) as control 2 min prior to infection[29].

**Quantification of fungal burdens in the blood and organs**. For fungal burden determination, mice were infected with $5 \times 10^6$ *C. neoformans* H99 or *C. albicans* and euthanized at various time points after infection. To determine fungal burden in the blood, mice were anesthetized by a ketamine/xylazine mixture; blood was taken by cardio-puncture using a 1 ml syringe containing 50 μl of heparin (100 U/ml). To determine fungal burdens in the tissues, the lung, liver, brain, kidney, and spleen were collected into 15 ml tubes each containing 2 ml ice cold PBS. To avoid contamination of tissues by circulating fungi, perfusion was performed using 20 ml PBS from the left ventricle after cutting open the inferior vena cava. The organs were homogenized with a hand-held tissue homogenizer (Omni international,

USA). After serial dilution and plating on Sabouraud dextrose agar plate, CFUs were enumerated after 36 h of growth at 32 °C.

**Intravital microscopy (IVM)**. Mice were anaesthetized with a mixture of ketamine (200 mg/kg) and xylazine (10 mg/kg). After anesthesia, cannulation of the tail vein was performed for injection of additional mixture of ketamine and xylazine to maintain anesthesia and for injection of yeast cells or other reagents. A midline laparotomy was performed by making a midline incision followed by a lateral incision along the costal margin to the midaxillary line. A high temperature cautery was used to prevent blood vessel from bleeding when cutting. The xiphoid process was lifted using a suture attached to it, and the connective tissues between the liver and the diaphragm was cut. The mice were placed in right lateral position. After cutting the connective tissues between the liver and stomach, the liver was externalized onto a customized acrylic imaging stage and covered with a cover slip. A heating lamp was used to maintain the body temperature of the mice throughout the imaging. The liver was continuously moistened with a saline-soaked sponge to prevent tissue dehydration and to help restrict movements. The heart rate was maintained constant during the period of imaging (Supplementary Fig. 8). The perfusion of the blood to the liver was monitored by visualization of the blood cell movement under the intravital microscope.

**Video capture and analysis**. The Zeiss Axio Examiner Z1 system (Zeiss, Germany) was used for the intravital imaging. After the preparations and settings, mice were infected with $100 \times 10^6$ GFP-expressing *C. neoformans* through the catheter in the tail vein and video recording began immediately. In some experiments, mice were i.v. injected with 2 μg of PE conjugated anti-F4/80 (Catalog No. 123110, Biolegend) or anti-PECAM-1 (Catalog No. 102508, Biolegend) antibodies to label liver KCs or sinusoids 5 min prior to injection of the yeast cells. A 20× objective lens was used throughout the experiment and time-lapse videos were recorded at the speed of 1 frame per second for later playback and analysis. Due to limited tissue penetration of visible light, a superficial field of view (FOV) of up to 40 μm was focused. One FOV was recorded per mouse. For statistical analysis, the following events were counted for each video: (1) the total number of yeast cells that appeared in the (FOV, 0.5 mm × 0.5 mm) within a time interval; (2) the number of yeast cells that were captured in the FOV within a time interval; and (3) the number of yeast cells which escaped after capture. We define capture as a fungus coming to a full stop and remaining stationary for at least 3 s. Those yeast cells staying in a locus for more than 3 s and being released into circulation were deemed as an escape. Next, we calculated the probability of fungi getting captured by comparing the number of captured yeast cells with total appearing yeast cells. The probability of escape was calculated by comparing the number of escaped yeast cells with total captured yeast cells. Fungal clearance in the blood (% blood clearance) was calculated as the number of free fungi that entered the FOV within a one-minute window divided by the number of free fungi that entered the FOV within the first minute (the first minute is defined as 0% clearance). The half-life of circulating fungi was calculated as the time needed to clear 50% of circulating fungi in the blood.

**Immunohistochemistry and confocal microscopy**. The tissues were removed and frozen in OCT compound. Frozen tissue blocks were cut on a cryostat microtome with a thickness set to 7 μm/section; and sections were placed on coated glass slides. Tissue sections were fixed in ice cold acetone for 10 min. After fixation, the samples were blocked by incubation with 5% goat serum for 30 min, followed by incubation with the primary antibody at 4 °C overnight. After three washes with PBS containing 0.05% Tween 20, sections were incubated for 30 min with the fluorescence conjugated second antibodies. After washing, the tissues were stained with DAPI to show the nuclei and persevered in mounting buffer. The tissue slides were examined under the Zeiss LSM 510 system. To confirm depletion of KCs, mice were i.v. injected with 2 μg Alex Fluor 488 anti-F4/80 mAb (Catalog No. MF48020, Invitrogen) and 2 μg Alex Fluor 647 anti-PECAM1 mAb (Catalog No. 102516, Biolegend) in 200 μl PBS to label KCs and blood vessels, respectively. Ten minutes later, mice were euthanized and perfused with 20 ml cold PBS after cutting the vena cava open until the liver became pale. Then the whole liver was excised carefully and placed on a 35-mm glass-bottom dish (thickness no. 1.5; MatTek, USA) containing 1 ml RPMI 1640 with 10% FBS. The whole-mount liver was examined under the Zeiss LSM 510 system using 63X oil immersion lens.

**Phagocytosis assay**. Mice were i.v. infected with $20 \times 10^6$ GFP-expressing *C. neoformans*. To label extracellular yeast cells, the infected mice were i.v. injected with 100 μl of Uvitex 2B (1% w/v in PBS, Polysciences Inc.) at various time points after infection. The staining of extracellular *C. neoformans* by Uvitex 2B occurred almost immediately after administration of Uvitex 2B. Mice were euthanized 2 min after treatment of Uvitex 2B and perfusion was performed; the liver was excised and examined under a microscope to count the number of yeast cells inside or outside phagocytes. *C. neoformans* inside or outside phagocytes can be readily distinguished by the negative or positive staining of Uvitex 2B. For each animal, 50 or more yeast cells were counted. The advantage of this method is that the in/out status of *C. neoformans* will be instantly marked after Uvitex 2B injection; even if the fungus later gets phagocytized, the staining at the time of Uvitex 2B treatment

will remain. In some experiments, phagocytosis of *C. neoformans* was confirmed using flow cytometry by distinguishing a double labeling of GFP and Uvitex 2B or a single labeling of Uvitex 2B.

**Growth assay**. To compare the fungal growth in different organs, mice were i.v. infected with $20 \times 10^6$ GFP-expressing *C. neoformans* or Uvitex 2B-labeled *C. albicans*. The infected mice were euthanized at various time points post infection; different organs were excised and immediately visualized under microscope. In some experiments, mice were i.v. injected with 200 μl of CLL or PBS liposomes as control to deplete KCs 24 h before infection. The total number of yeast cells in each colony (each cluster of fungi was considered to be one colony) were counted under a microscope. A colony containing at least two yeast cells was defined as growing into a cluster.

**Elisa assay**. The level of IFN-γ in the liver homogenate was determined using IFN-γ Elisa Kit (BD Biosciences) following the manufacturer's instructions. After euthanasia, the liver was collected and homogenized in a 15 ml tube containing 2 ml cold PBS, the supernatant was collected for Elisa after centrifugation at $3000 \times g$ for 10 min.

**Statistics**. GraphPad 5.0 was used for all statistical analyses. For single comparisons, an unpaired two-sided Student's *t* test was performed; for multiple comparisons, one-way analysis of variance (ANOVA) followed by Tukey's post hoc test or two-way ANOVA was used. In both cases, a *p* value of <0.05 is considered significant.

**Reporting summary**. Further information on research design is available in the Nature Research Reporting Summary linked to this article.

## Data availability

The datasets generated during and/or analyzed during the current study are available from the corresponding author on reasonable request. The source data underlying Figs. 1c, 2c–i, 3b–g, 4c–e, g–i, 5b, d–g, 6b–d, f–j and 7a, b, d, f and Supplementary Figs. 1, 2b, 3, 4a, b, 5b–d, f, 7c and 8 are provided as a Source Data file.

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

## Acknowledgements

We thank Kenneth Class (MPRI Flow Cytometry Core Facility, University of Maryland) for his assistance with FACS analyses and Dr. Yunsheng Wang (Department of Veterinary Medicine, University of Maryland) for his assistance with confocal imaging. Chinese Scholarship Council (CSC) provided fellowship to Peng Sun, Hongmei Li, Gongguan Liu, and Yong Fu. U.S. - Egypt Science and Technology Joint Fund (STDF) provided fellowship to Mohammed Yosri. National Institutes of Health (NIH) provided funding to Meiqing Shi under grant number AI131219 and AI131905.

## Author contributions

D.S. designed the overall study, performed experiments, performed data analyses, and wrote the manuscript; P.S. performed experiments; H.L., M.Z., G.L., A.S., Y.C., Y.F. J.X., and M.Y. assisted in the experiments; Y.N., H.Z., and X.Z. provided discussion and revised the manuscript; and M.S. supervised the study and wrote the manuscript.

## Competing interests

The authors declare no competing interests.
