## [Peer Review File · Nature Communications]

Reviewers' comments:

Reviewer #1 (Remarks to the Author):

The authors report effects of Kupffer cells and specific receptors on these cells on clearance of fungal pathogens, particularly *C. neoformans*. *C. neoformans* is a medically important pathogen in immunosuppressed persons, making this a medically relevant area of investigation that will be of interest to scientists and physician-scientists in infectious disease, immunology, and imaging. The manuscript is well-written with conclusions supported by presented data. Use of intravital microscopy to analyze pathophysiology of disease and results of various interventions is innovative. Overall use of statistics is appropriate, although the manuscript lacks power calculations or justification for numbers of mice used in experiments. The authors need to provide additional details about the imaging methods to interpret presented data, particularly since the major conclusions rely heavily on in vivo imaging of liver. Specific comments are listed below.

1. The imaging time period appears to be 50 minutes or less. The authors should comment on the rationale for this time period. In this context, the authors should report if the physiology of the mouse (heart rate, perfusion to liver) remain constant during the time period of surgery and imaging and how they monitored these parameters.
2. The manuscript should report at the depth at which they acquired images in liver. Given that they used visible laser lines, images likely are from very superficial sites in the liver. Did the research group investigate deeper sites in the liver to ensure that observations are representative for the entire organ?
3. Figure 1. Panel C reports the number of fungi captured. The authors should report the number of total events. The manuscript also does not report numbers of fields of view sampled per imaging session per mouse. These comments apply to other figures reporting data from intravital microscopy.

Reviewer #2 (Remarks to the Author):

This is an interesting and unusual paper that reports a role for liver phagocytes (Kupffer cells) in the clearance of the fungal pathogen *Cryptococcus neoformans*. Relatively few studies have examined

the role of the liver in the aetiology of this disease and this manuscript makes use of some very impressive intravital imaging to examine this for the first time. The authors show convincingly that's circulating critical cells become trapped within the liver vasculature. This trapping relies on phagocyte function, since clodronate depletion eliminates the capture. Most importantly, when live capture is inhibited, circulating fungal burden increases - suggesting that trapping and elimination of the pathogen in the liver may be important in reducing organ dissemination.

Overall, I think this work represents an important and interesting development for the field. However there is one major concern, which is that clodronate depletion is not specific for Kupffer cells. Indeed previous authors have used the same method to deplete circulating monocytes and shown the reverse of the findings here – ie circulating monocytes are important to traffic Cryptococci into the brain and thus clodronate depletion improves survival (Charlier et al, 2009).

These two findings are not necessarily contradictory, since phagocytes may play an important role in clearance early on in the infection but an equally important role in Trojan horse trafficking later in the disease. However, it is incumbent on these authors to demonstrate the specificity of the effect they see. To that end I would recommend the authors repeat the main focus of this work but using an alternative Kupffer-cell depletion method. I recognise that highly-specific KC depletion methods do not yet exist, but it would be a significant advantage if they were to use a second method that has alternative 'off-target' effects; for instance, gadolinium chloride depletion of KCs, and/or by doing the reverse approach and using a transgenic method to remove circulating phagocytes but not Kupffer cells. Alternatively, might the specific expression of CR1g, described by the authors, be a means by which they could specifically deplete KCs?

Two more minor points should also be addressed:

- a) If liver 'trapping' is important for cryptococcal clearance, then one might imagine that patients with cirrhosis or hepatitis might show enhanced susceptibility to cryptococcal disease – is data to test this readily available?
- b) The manuscript suffers from poor grammar throughout (e.g. line 151, "decrease of liver CFU in C3-/- mice was not due to the initial stop of *C. neoformans*. In fact, there was no much difference in capture probability") and should be proof-read carefully before resubmission.

Response to Reviewers

Reviewer #1 (Remarks to the Author):

The authors report effects of Kupffer cells and specific receptors on these cells on clearance of fungal pathogens, particularly C. neoformans. C. neoformans is a medically important pathogen in immunosuppressed persons, making this a medically relevant area of investigation that will be of interest to scientists and physician-scientists in infectious disease, immunology, and imaging. The manuscript is well-written with conclusions supported by presented data. Use of intravital microscopy to analyze pathophysiology of disease and results of various interventions is innovative. Overall use of statistics is appropriate, although the manuscript lacks power calculations or justification for numbers of mice used in experiments. The authors need to provide additional details about the imaging methods to interpret presented data, particularly since the major conclusions rely heavily on in vivo imaging of liver. Specific comments are listed below.

We thank the reviewer for the compliments.

As suggested, we have provided additional details about the imaging methods to interpret the presented data in the revised version. For example, we have explained the way to label KCs and sinusoids (line 480-482), the way to monitor the liver perfusion by visualization of the blood cell movement (line 473-474), the way to calculate the percentage of blood clearance and the half-life of the circulating yeast cells (line 494-499), the mouse heart rate (line 472-473), the depth of imaging (line 484-485), the numbers of fields of view per mouse (line 485), etc. either in “materials and methods” or in “figure legends” (also see below).

1. The imaging time period appears to be 50 minutes or less. The authors should comment on the rationale for this time period. In this context, the authors should report if the physiology of the mouse (heart rate, perfusion to liver) remain constant during the time period of surgery and imaging and how they monitored these parameters.

We selected this time period based on our published observations that the number of *C. neoformans* passing through brain postcapillaries was dramatically reduced within 30 min and that *C. neoformans* was hardly seen passing through the vasculature at 60 min (Shi et al. 2010 Journal of Clinical Investigation 120: 1683–1693. PMID: 20424328). To further provide the rationale, we returned to the lab and examined the kinetics of blood CFU in mice infected with *C. neoformans*. As shown in the **new Supplementary Figure 1**, 99% of the yeast cells were cleared from the blood 30 min after infection. We add the new data on line 106-107.

Surgical preparation and liver imaging were based on methods described previously by other groups (Geissmann et al. 2005 PLOS Biology 3:e113. PMID: 15799695; Lee et al. 2010 Nature Immunology 11:295-302. PMID: 20228796; Wong et al. 2011 Science 334:101-105. PMID: 21921158; Zeng et al. 2016 Cell Host & Microbe 20:99-106. PMID: 27345697). It seems that these groups did not mention the measurement of the heart rate and perfusion to

liver during intravital imaging on the liver. We followed these groups for the experimental procedure and did not measure the heart rate during the in vivo imaging. However, we understand the reviewer's concerns and agree that the heart rate is of importance. Thus, we purchased a heart rate monitor from Kent Scientific and examined the heart rate of mice during the period of intravital imaging using our standard protocol for anesthesia. As shown in the **new Supplementary Figure 8**, the heart rate remains relatively stable during the experimental procedure under our standard protocol (line 472-473). Regarding the perfusion to liver, we are able to visualize the blood cell movement under the intravital microscope. Thus, the perfusion of the blood to the liver was monitored by visualization of the blood cell movement under the intravital microscope. We have added this information to the "materials and methods" in the revised manuscript (line 473-474).

2. *The manuscript should report at the depth at which they acquired images in liver. Given that they used visible laser lines, images likely are from very superficial sites in the liver. Did the research group investigate deeper sites in the liver to ensure that observations are representative for the entire organ?*

Like every technique, intravital imaging has both strengths and limitations. Due to limited tissue penetration of the intravital microscope, a superficial field of view of up to 40 μm was focused. We have added this information to the "material and methods" in the revised manuscript (line 484-485). We agree with the reviewer's comment that images are from superficial sites in the liver. Therefore, we returned to the lab and collected data on yeast capture from liver frozen sections which reflected events occurring in deeper sites of the liver. The new data have been added in **new Supplementary Figure 2B** of the revised version and added in the revised manuscript (line 135-136). In addition, the CFU data of the entire liver tissue shown in the original figures serve as complimentary evidence to support the findings by intravital microscopy.

3. *Figure 1. Panel C reports the number of fungi captured. The authors should report the number of total events. The manuscript also does not report numbers of fields of view sampled per imaging session per mouse. These comments apply to other figures reporting data from intravital microscopy.*

We agree. We reanalyzed the videos and included the events of free yeast cells in **new Figure 1C**. Thus, the new Figure 1C showed the number of total events including both captured and free yeast cells in the revised manuscript (Line 769-772).

In order to analyze the kinetics of yeast capture in a period of time, one field of view was recorded per mouse. Statistics were made from the data of multiple mice. This is because the intravital microscope can only focus on one field of view and the technique does not allow us to record multiple fields of view simultaneously during a period of time. We have added this information in the "materials and methods" of the revised version (line 485).

Reviewer #2 (Remarks to the Author):

This is an interesting and unusual paper that reports a role for liver phagocytes (Kupffer cells) in the clearance of the fungal pathogen Cryptococcus neoformans. Relatively few studies have examined the role of the liver in the aetiology of this disease and this manuscript makes use of some very impressive intravital imaging to examine this for the first time. The authors show convincingly that circulating critical cells become trapped within the liver vasculature. This trapping relies on phagocyte function, since clodronate depletion eliminates the capture. Most importantly, when live capture is inhibited, circulating fungal burden increases - suggesting that trapping and elimination of the pathogen in the liver may be important in reducing organ dissemination.

We thank the reviewer for the support and encouragement.

Overall, I think this work represents an important and interesting development for the field. However there is one major concern, which is that clodronate depletion is not specific for Kupffer cells. Indeed previous authors have used the same method to deplete circulating monocytes and shown the reverse of the findings here – ie circulating monocytes are important to traffic Cryptococci into the brain and thus clodronate depletion improves survival (Charlier et al, 2009).

These two findings are not necessarily contradictory, since phagocytes may play an important role in clearance early on in the infection but an equally important role in Trojan horse trafficking later in the disease. However, it is incumbent on these authors to demonstrate the specificity of the effect they see. To that end I would recommend the authors repeat the main focus of this work but using an alternative Kupffer-cell depletion method. I recognise that highly-specific KC depletion methods do not yet exist, but it would be a significant advantage if they were to use a second method that has alternative ‘off-target’ effects; for instance, gadolinium chloride depletion of KCs, and/or by doing the reverse approach and using a transgenic method to remove circulating phagocytes but not Kupffer cells. Alternatively, might the specific expression of CRIg, described by the authors, be a means by which they could specifically deplete KCs?

We agree with the reviewer’s explanation that phagocytes (notably KCs) play an important role in clearance early on in the infection but an equally important role (notably monocytes) in Trojan horse trafficking later in the disease.

As suggested by the reviewer, we returned to the lab and examined the liver capture of *C. neoformans* using an alternative KC depletion method, i.e. treatment with gadolinium chloride. As shown in the **new Supplementary Figure 3**, treatment of mice with gadolinium chloride significantly reduced liver fungal burden, resulting in higher fungal burdens in the blood and other tissues (line 146-148).

Clodronate liposome depletes KCs and monocytes but not neutrophils and dendritic cells (Van Rooijen et al. 1994 Journal of immunological methods 174: 83-93. PMID: 8083541; Ferenbach et al. 2012 Kidney international 82: 928-933. PMID: 22673886). Monocytes exist in two major populations, termed Ly6C^{hi} and Ly6C^{low} monocytes. It is known that CCR2^{-/-} mice and Nur77^{-/-} mice lost Ly6C^{hi} subset and Ly6C^{low} subset, respectively. It has been recently shown that treatment of Nur77^{-/-} mice with anti-CCR2 mAb depleted Ly6C^{hi} monocytes in Nur77^{-/-} mice (Michaud et al. 2013 Cell Reports 5:646-653. PMID: 24210819).

To rule out the potential interference of monocytes, we infected CCR2^{-/-} mice (loss of circulating Ly6C^{hi} monocytes), Nur77^{-/-} mice (loss of Ly6C^{low} monocytes) and Nur77^{-/-} mice treated with anti-CCR2 mAb (loss of both Ly6C^{hi} and Ly6C^{low} monocytes) with *C. neoformans*, as suggested by the reviewer. As shown in the **new Supplementary Figure 4**, there was no significant difference in the fungal burdens in the liver and other tissues among those groups of mice 3 h after infection, demonstrating that loss of Ly6C^{hi} and/or Ly6C^{low} monocytes does not affect the capture of *C. neoformans* in the liver (line 148-153).

Two more minor points should also be addressed:

a) *If liver ‘trapping’ is important for cryptococcal clearance, then one might imagine that patients with cirrhosis or hepatitis might show enhanced susceptibility to cryptococcal disease – is data to test this readily available?*

Unfortunately, we do not have the chance to study clinical cases. However, accumulating clinical studies indicated that patients with cirrhosis or end-stage liver diseases are more susceptible to brain infections with *C. neoformans*, establishing a link between liver diseases and the enhanced risk to cryptococcal meningoencephalitis (e.g. PMID: 25747471; PMID: 25806406; PMID: 26835475).

b) *The manuscript suffers from poor grammar throughout (e.g. line 151, “decrease of liver CFU in C3-/- mice was not due to the initial stop of C. neoformans. In fact, there was no much difference in capture probability”) and should be proof-read carefully before resubmission.*

We apologize for the existence of grammatical mistakes, and have carefully proof-read the manuscript in the revised version.

REVIEWERS' COMMENTS:

Reviewer #1 (Remarks to the Author):

The authors appropriately addressed comments from the review.

Reviewer #2 (Remarks to the Author):

The authors have comprehensively addressed my major comment, regarding the specificity of the effect to Kupffer cells. By using circulating monocyte depletion strategies, and a more specific KC depletion approach, they add substantial new data which strongly supports their original hypothesis. Together, I think this makes this paper a very solid and important contribution to the literature. I have no further criticisms of the work, which now makes for a very exciting and interesting read.

Reviewer #1 (Remarks to the Author):

The authors appropriately addressed comments from the review.

We appreciate the reviewer for the support.

Reviewer #2 (Remarks to the Author):

The authors have comprehensively addressed my major comment, regarding the specificity of the effect to Kupffer cells. By using circulating monocyte depletion strategies, and a more specific KC depletion approach, they add substantial new data which strongly supports their original hypothesis. Together, I think this makes this paper a very solid and important contribution to the literature. I have no further criticisms of the work, which now makes for a very exciting and interesting read.

We are grateful for the encouraging comments and appreciate the reviewer for the support.